# Eating Disorders and Gastrointestinal Diseases

**DOI:** 10.3390/nu11123038

**Published:** 2019-12-12

**Authors:** Antonella Santonicola, Mario Gagliardi, Michele Pier Luca Guarino, Monica Siniscalchi, Carolina Ciacci, Paola Iovino

**Affiliations:** 1Department of Medicine, Surgery and Dentistry, “Scuola Medica Salernitana”, University of Salerno, 84084 Salerno, Italy; antonellasantonicola83@gmail.com (A.S.); mariogagliardi@outlook.com (M.G.); msiniscalchi@unisa.it (M.S.); cciacci@unisa.it (C.C.); 2Unit of Gastroenterology, Campus Bio Medico University, 00128 Rome, Italy; m.guarino@unicampus.it

**Keywords:** anorexia nervosa, bulimia nervosa, binge-eating disorders, gastrointestinal diseases

## Abstract

Eating disorders (ED) are frequently associated with a wide range of psychiatric or somatic comorbidities. The most relevant ED are anorexia nervosa (AN), bulimia nervosa (BN), and binge eating disorders (BED). Patients with ED exhibit both upper and lower gastrointestinal (GI) symptoms. Evidence of alterations throughout the GI tract in ED will be analyzed given the role of the GI tract in food intake and its regulation. It remains a matter of debate whether GI disorders are inherent manifestations of ED or the results of malnutrition occurring from ED. Moreover, recent clinical studies have highlighted the growing role of intestinal microbiota in the pathogenesis of ED, making it possible to hypothesize a modulation of intestinal microbiota as a co-adjuvant to standard therapy. The aim of this review is to analyze the link between ED and GI diseases and to present, where known, the potential key factors underlying these conditions. Conclusions: The presence of GI disorders should be investigated in patients with ED. Screening for ED should also be encouraged in individuals seeking treatment for unexplained GI complaints to better address therapeutic issues that surround these difficult medical conditions.

## 1. Eating Disorders

Eating disorders (ED) are a crucial health problem in Western countries and mainly in the young population [1]. ED are very complex, lacking a pathophysiological definition, and are clinically recognized by strict diagnostic criteria of the Fifth Edition of the Diagnostic and Statistical Manual of Mental Disorders (DSM-V), released by the American Psychiatric Association [2]. Anorexia nervosa (AN), Bulimia nervosa (BN) and Binge Eating Disorder (BED) are the three major categories of ED (Figure 1).

ED can be associated with anxiety or mood disorders; insomnia; as well as somatic complications such as cardiac alterations, hormonal imbalances and also a wide range of gastrointestinal (GI) symptoms [3].

Prevalence estimate of lifetime AN is 0.8% [4]. The overall incidence rate of AN is around 7 per 100,000 persons; however, it is higher in 15–19 year old girls reaching 109.2 cases per 100,000. The incidence is increasing in Italy and other Western countries probably due to earlier diagnosis or earlier onset [5,6].

The prevalence of BN is 0.28% [4]. The highest incidence rate of BN is 300 per 100,000 persons in women aged between 16 and 20 [7], although the age at onset is decreasing [5].

BED are described in 0.85% of the population [4,8] and are frequently associated with medical comorbidities, especially obesity and its complications such as Type II diabetes mellitus, hypertension, and dyslipidemia [9]. Furthermore, BED may occur in a subset ranging from 27% to 47% in severely obese persons undergoing bariatric surgery [10].

The previous edition of DSM (DSM-IV) [11], classified as “Eating Disorders Not Otherwise Specified (EDNOS)”, is the ED that does not satisfy all the diagnostic criteria for AN or BN. Previously, EDNOS contained a huge and varied group of individuals and was the most commonly diagnosed ED. However, the DSM-V revised this ED type and called it “Other Specified Feeding or Eating Disorder” (OSFED). Moreover, the DSM-V introduced the definition of Avoidant/restrictive food intake disorder (ARFID) that was distinct from AN and BN in that there is no body image disturbance [2].

The first description of AN in the 19th century revealed that the sufferers’ reasons for self-starvation generally were strictly connected to inappetence, feelings of fullness and abdominal pain or constipation [12,13]. Since then, a wide range of GI symptoms in ED patients has been described [14].

More recently, a prospective study reported that the majority of ED patients complained of postprandial fullness (96%) and abdominal distention (90%) [15], and pooled GI symptoms were significantly correlated with hypochondriasis, producing a deleterious influence of psychiatric comorbidities on GI discomfort [15].

BED co-occurs with a variety of psychiatric disorders, especially mood (54.2%), anxiety (37.1%) and substance use disorders (24.8%) [16]. The prevalence of psychiatric comorbidities in women with BED is higher compared to those with AN and BN [17].

Furthermore, in ED patients a significant alteration of physical health and psychosocial functioning has been described [18].

## 2. ED and GI Tract

The underlying mechanisms that could explain the link between ED and the GI tract are numerous and only partially understood. It has been suggested that ED behavior including vomiting, laxative abuse, and restrictive food intake has a strong impact on the function of the GI tract; conversely, disablement of the GI function such as disordered motility can bolster typical symptoms of ED such as loss of appetite, self-induced vomiting, dysphagia, constipation, and bloating [15,19]. In addition, the malnutrition occurring in ED [20], producing metabolic myopathy, as well as electrolyte depletion, seems to dramatically influence the impairment in gastric motility, gastric emptying and intestinal transit [21].

However, it is noteworthy that several studies have excluded the presence of structural or metabolic GI disorders in the majority of ED patients [22,23,24,25,26,27]. GI symptoms in ED patients usually reflect an impaired GI function [14]. Conversely, a structural illness like celiac disease may increase the likelihood of developing an ED because the patient pays attention to food intake, body weight and food-related GI symptoms that may result in chronic dietary restriction [28].

The arrival of food in the GI tract produces a multiplicity of signals rising from gastric distension, nutrient and non-nutrient elements, and also gut hormones released from the enteroendocrine cells of the GI wall [29,30,31].

The gradual distension of the gastric wall to accommodate food gives rise through the stimulation of the mechano-sensitive vagal afferent fibers [32,33], to the perception of fullness; this plays a central role in controlling meal size [34,35]. As gastric emptying proceeds, the signals rising from mechanical distension reduce gradually, and it moves to the antrum and then progresses into the small intestine. The antrum has a crucial role in the perception of fullness and meal termination [36]. Simultaneously, nutrients influence postprandial GI functions, such as GI motility and GI hormone release, with effects on appetite perceptions and consequent energy intake linking central perceptions with the GI function [37]. Dysregulations of these physiological processes can adversely affect eating behavior or provoke GI symptoms, such as exaggerated postprandial fullness, nausea and bloating [38,39].

Considering the vastness of the issue, we do not cover triggering factors (e.g., stress, emotions) and other comorbid conditions (anxiety, depression, addiction) that may affect at the same time eating behavior and digestive function. This topic was described elsewhere [40,41].

In this review, the evidence of alterations throughout the GI tract will be analyzed, taking into account the role of the GI tract as the organ system associated with food intake and its regulation. Similarly to Rome Criteria, clinical features were identified within each region of the GI tract [42]. Moreover, it is a matter of debate if GI symptoms are clinical manifestations correlated to ED characteristics, for example, ED type [20].

Thus, our aim was to systematically describe GI involvement from esophagus to intestine in the three main ED disorders, AN, BN and BED in an attempt to offer a useful tool to practicing physicians in the recognition and, possibly, in planning the management of GI alterations in these difficult patients.

A special section was dedicated to Gut Microbiota, which could be considered a “new organ” of the digestive system.

## 3. Esophagus

### 3.1. Anorexia Nervosa

Heartburn, non-cardiac chest pain, dysphagia, and globus are esophageal symptoms that raise the suspicion of esophageal disorders. These symptoms are frequently observed in patients with ED [43].

One of the most cited studies was performed on 30 AN patients decades ago. It found a very high percentage of them suffered from esophageal motility disorders such as achalasia (23%) or other esophageal motility abnormalities (27%), suggesting that esophageal dysmotilities can be mistakenly diagnosed as AN [44]. A more recent study [45] demonstrated that the presence and severity of the three symptoms considered to be esophageal in origin (dysphagia, heartburn and regurgitation), were significantly higher in restrictive and binge-eating/purging types of AN patients compared to controls, with no correlation with esophageal motor disturbances and no improvement after rehabilitation. Specifically, using an eight-channel catheter, no abnormalities were shown in esophageal manometry tracings, only a higher LES tone in restricting AN patients that returned to normal after rehabilitation; post-deglutitive relaxations were always complete, and the motility of the esophageal body was substantially normal apart from a single patient that fulfilled the manometric diagnosis of nutcracker esophagus according to Castell et al. [46]. These results could suggest that either AN can overlap with esophageal dysfunction or esophageal dysfunction can mimic AN symptoms [44]. To date, there are not available data about novel technologies, such as High Resolution Manometry (HRM), that possibly allow a different diagnosis on the basis of the new Chicago Classification [47] in these patients.

It has been suggested that self-induced vomiting in patients with purging type AN could potentially cause esophageal acidic damage similar to that occurring in gastroesophageal reflux disease (GERD), increasing the risk of developing sequelae such as Barrett’s esophagus [48] and adenocarcinoma. Although in a systematic review and meta-analysis AN was potentially associated with an increased risk of developing esophageal cancer (two studies in women; RR, 6.10; 95% CI, 2.30–16.18; *p* < 0.001; *I*^2^, 0%) [49], the evidence was judged to be of low confidence and there is still uncertainty about the associations and the role of modifiable risk factors such as smoking.

### 3.2. Bulimia Nervosa

An association between heartburn and regurgitation, typical GERD symptoms, is suggested as possible in BN patients, since self-induced vomiting is a recurrent inappropriate behavior in these patients, but is far from conclusive [50]. Esophagitis has been reported in some patients with BN; however, Kiss et al. suggested that chronic and self-induced vomiting rarely leads to mucosal injury [51]. Several case reports described Barrett’s and carcinoma of the esophagus in patients with a prior history of BN or psychogenic vomiting [52,53]. Conversely, a recent study demonstrated that patients with ED had an increased risk of developing esophageal cancer but no cases were adenocarcinomas. Brewster et al. hypothesized that alcohol, smoking, and nutritional deficiency could play more of a role than acidic damage through self-induced vomiting [54].

An alteration in esophageal motility was described in an early study in which 8/26 patients with BN had motor abnormalities and five of them fulfilled diagnosis for achalasia and DES, demonstrating the need to take a thorough history regarding swallowing and vomiting in these patients [55], which is a finding unconfirmed by others [56].

Lastly, bleeding can also occur as a result of tears in the esophagus (Mallory–Weiss Syndrome) due to vomiting [57].

### 3.3. Binge Eating Disorder

Cremonini et al. [58] demonstrated that BED was significantly associated with heartburn (OR 2.2, 95% CI 1.4–3.3), acid regurgitation (OR 2.3, 95% CI 1.5–3.6), and dysphagia (OR 3.3, 95% CI 1.6–6.8). The huge food intake during a binge episode seems to alter the lower esophageal sphincter’s function, influencing the amount of gastric refluxate that reaches the distal esophagus and leading to heartburn and regurgitation [58,59]. The acid reflux could promote the onset of dysphagia in these patients [59,60]. Conversely, other authors found a similar prevalence of functional esophageal disorders in BED patients and those affected by other eating disorders [43].

We previously reported that the frequency-intensity scores of typical and atypical GERD symptoms did not differ in obese patients with and without a binge eating behavior pattern [61], although it has been extensively demonstrated that obese patients have a high prevalence of GERD [62].

## 4. Stomach

### 4.1. Anorexia Nervosa

Stacher proposed that AN patients commonly complain of GI symptoms, subtly implying a disordered gastric motility, especially when they are in a refeeding phase [63]. Moreover, AN reported notably more fullness and less hunger than healthy controls [64]. In addition, dyspeptic symptoms may serve as an argument for food refusal [65]. We previously demonstrated that AN patients had an increase in postprandial fullness score and prominent early satiety, fulfilling the diagnosis of Postprandial Distress Syndrome (PDS) according to Rome III Criteria [66]. To analyze the key factors underlying the origin of these symptoms, numerous studies on gastric motility have been performed.

AN patients have shown slow gastric emptying for liquids [24,67,68] and solids [22,27] using scintigraphic scanning techniques in both restricting and binge/purging AN subtypes [69]. Some authors demonstrated that GI symptoms such as nausea, vomiting, and gastric fullness correlated with slow gastric emptying [25], whereas satiety did not show any correlation [70]. Others did not find any correlation among gastric emptying, dyspeptic symptoms and body weight in AN [24,69]. These discordant results between gastric emptying and symptoms suggest that GI symptoms are unreliable indicators of gastric emptying in patients with ED and further studies are needed in this field. Noteworthy is that gastric half emptying time studied by scintigraphy showed a negative correlation with BMI [64]. Gastric emptying and symptoms improved following weight rehabilitation [64,71], even without reaching normal BMI [72], and in an independent manner, whereas short-term refeeding did not [69]. From these findings, it is not recommended to perform routine gastric-emptying studies for the diagnosis of gastroparesis because it is expected to worsen as the severity of weight loss increases.

### 4.2. Bulimia Nervosa

Patients with BN and EDNOS complained of postprandial fullness, epigastric pressure and nausea as their most frequent and bothersome dyspeptic symptoms [66]. In BN patients, the huge amount of food intake during a binge not only leads to a feeling of loss of control but also to a perception of epigastric distension [66]. The latter, together with epigastric pain that is commonly associated, is concluded by self-induced vomiting, which allows either continuation or termination of the binge [63].

Previous studies in BN patients focused their attention on gastric emptying and hypothesized that delayed gastric emptying might contribute to decreased postprandial CCK release and a blunt subjective sense of satiety, leading to uncontrolled overconsumption and possibly to compensatory purging [73]. Other studies demonstrated that gastric emptying of solids and liquids was similar to healthy subjects [24,68,74]. Beyond delayed gastric emptying, other mechanisms such as an increased gastric capacity and a diminished sensitivity to gastric distention may play a role [75,76]. In lean individuals, an increased activation of the amygdala was shown in those with a lower BMI, suggesting a smaller gastric capacity [77].

In addition, disturbance in peripheral hormone secretion has been described in AN and BN patients although data are conflicting. ED patients, especially AN patients, showed increased fasting levels of PYY and CCK; however, postprandial responses of PYY and CCK do not show consistent behavior [78]. AN but not BN patients had increased fasting ghrelin levels. Conversely, postprandial suppression of ghrelin is reduced in BN but not in AN. Authors suggested that higher ghrelin levels in ED could be an effort to compensate for the state of malnutrition of these patients. In fact, BN patients with normal BMI showed similar levels of ghrelin compared to healthy controls; while in AN patients, ghrelin levels return to normal after weight regain. However, it is still unclear if this dysregulation is a consequence of the nutritional changes in ED [78].

Though AN can be complicated by a superior mesenteric artery syndrome due to duodenal obstruction [79], only one case was described for BN [80].

### 4.3. Binge Eating Disorders

Crowell et al. [81] demonstrated that upper GI symptoms were more common in obese binge eaters than normal controls or obese non-binge eaters. The specific pattern of eating behavior may contribute to the development of these symptoms [61].

Geliebter et al. demonstrated that BED subjects have a larger gastric capacity than non-BED subjects, hypothesizing that it could reduce satiation following a fixed-size meal and contribute to consumption of a binge meal [82]. Other authors [83] confirmed these findings, showing that BED patients consumed a significantly higher quantity of food than obese and normal-weight controls. They also observed that BED patients showed an alteration in self-reported hunger and fullness, especially towards the end of the meals, in contrast to obese and normal-weight controls. This could cause a reduced response to the signals that lead to meal termination, allowing these patients to continue the binge episodes. Accordingly, PDS was significantly more frequent in OB with a Binge-Eating (BE) behavior pattern compared to OB without a BE behavior pattern [61]. From a physiological point of view, a huge food intake over a relatively short time could potentially impair the functional accommodation and emptying, contributing to the genesis of GI symptoms in obese individuals [39].

Geliebter et al. [82] reported a normal gastric emptying rate after the fixed meal, which is confirmed by normal CCK levels in BED patients, unlike that observed in BN. Thus, the specific pattern of eating behavior could play a role in the development of functional dyspepsia (FD) symptoms. Moreover, the increased gastric capacity was associated with coexistent lower gastrointestinal symptoms: The high amount of food in fact, may cause colonic discomfort due to the increased osmotic load downstream [84].

Cremonini, in addition, demonstrated that BED was significantly associated with bloating (OR 3.6, 95% CI 2.3–5.5) and upper abdominal pain (OR 2.3, 95% CI 1.4–3.6) [58].

Some case reports reported fatal outcomes caused by the extreme acute gastric dilation after a binge episode [85,86,87,88]. It has been hypothesized that the massive food intake might cause an acute gastric dilatation, compressing the gastric wall and producing necrosis and perforation of the stomach [87]. Kim et al. [89] reported a case of a 26-year-old woman who developed after a binge episode a huge gastric distension associated with acute pancreatitis. The authors hypothesized that the dilated stomach might cause a duodenal compression, increasing the intraduodenal pressure. This mechanism provoked the passage of duodenal contents retrogradely along the pancreatic duct, leading to the onset of pancreatitis [89].

## 5. Intestine

### 5.1. Anorexia Nervosa

Boyd et al. showed that 98% of ED patients, whom 44% AN patients, met Rome II criteria for no less than one Functional Gastrointestinal Disorders (FGIDs) and the most prevalent was IBS [14]. IBS is a functional GI disorder that is frequently associated with psychosocial impairment [90,91].

Moreover, Boyd et al. [14] suggested that somatization, neuroticism, and state and trait anxiety might be considered as predictors of specific FGIDs.

More recently, Sileri et al. [92] conducted a study on 83 anorectic patients, revealing defecatory disorders in 93% of them. Moreover, the results were strictly correlated with severity and duration of disease. Specifically, the defecatory disorders increased from 75 to 100% when BMI was less than 18 kg/m^2^ and from 60 to 75% when illness duration was longer than 5 years. Several studies have suggested a relation between constipation in AN and a delayed colonic transit studied using a radiopaque marker technique [71,93]. Chronic constipation is a frequent and extremely burdensome disorder negatively impacting social life and quality of life, with a higher prevalence among females [90,94]. It has been suggested that in ED, chronic constipation might be the result of abnormal colonic function due to poor or substandard intake of food and electrolytic alterations secondary to purging behavior, especially laxative abuse. Furthermore, it should be considered that constipation may be caused by antidepressant medications, particularly tricyclic antidepressants [20,95,96]. Another pathologic feature of anorectic patients, firstly evaluated in the aforementioned study by Chiaroni [93], is pelvic floor dysfunction. Indeed, the authors, using anorectal manometry, revealed that 40% of their AN population had pelvic floor dysfunction. In the same study, the sample with the dysfunction after a 4-week refeeding program showed a colonic transit within normal parameters but no improvement in the pelvic function, suggesting structural damage of the puborectalis muscle slings [92,93].

The correlation between pelvic dysfunction and a long-history of ED could be explained as follows: protracted evacuation efforts, purging laxative abuse, excessive exercise, and repeated forced vomiting induce an abnormal intra-abdominal pressure rise that may lead to structural damage of pelvic floor muscles in addition to atrophy and rhabdomyolysis due to malnutrition [97,98,99,100].

### 5.2. Bulimia Nervosa

Dejong found that 68.8% of BN patients had IBS according to Manning criteria [101]. Among bowel symptoms, constipation is the most frequent even in BN patients.

Whole-gut transit and mouth-to-cecum transit time were tested in anorectic and bulimic patients, using the radiopaque marker technique and the lactulose breath test, respectively [23]. Both values were significantly lower in BN patients than healthy controls in 67% of the cases. Other studies revealed a similar percentage of constipation in bulimic patients [14,102]. It has been suggested that the pathogenesis of constipation in bulimic patients does not differ from the anorectic patients, probably being the result of poor nutrition, electrolytic disturbances due to laxative abuse and the iatrogenic effect of antidepressant medications [20,95,96,103].

Moreover, a relationship has been hypothesized between BN and Rectal Prolapse (RP). The majority of BN patients use forced vomiting as a main purging method. During the process of vomiting, a deep breathing precedes the actual vomiting to protect the lungs from aspiration followed by a strong contraction of the diaphragm down with all the abdominal muscles to expel the contents of the stomach [104]. This repetitive intra-abdominal pressure in addition to constipation, pelvic floor weakness and over-exercise is considered the rationale for the relationship between BN and RP [100,105,106]. As previously mentioned, other common purging methods in bulimic patients are the laxative and enema abuse detected in 36.5% and 7.9% of patients with ED, respectively [107]. This purging behaviour could lead to several GI disorders. First, chronic watery diarrhea leading to dehydration and different electrolyte imbalances such as hypokalemia and metabolic acidosis with higher risk of fatal arrhythmias [108,109]. Second, melanosis coli, defined as a benign darkening of colonic mucosa due to chronic abuse of a specific type of laxative, gradually resolvable with laxative withdrawal [110]. Third the “cathartic colon” characterized by abnormal peristalsis due to continuous usage of stimulant laxatives; actually, this condition is rarely reported [111].

Despite the fact BN patients often report IBS-attributable symptoms, there are no studies evaluating this condition in ED using the new Rome IV criteria. For example, Dejong et al. [101] used Manning Criteria for estimating the prevalence of IBS in BN and EDNOS-BN (68.8%), and Boyd et al. [14] diagnosed IBS according to Rome II criteria. Future studies overcoming this limitation would need to define significantly the prevalence of IBS in ED.

### 5.3. Binge Eating Disorders

BED showed a high frequency of lower GI symptoms such as diarrhea (OR 3.1, 95% CI 2.0–4.6), urgency (OR 3.9, 95% CI 2.6–6.1), constipation (OR 2.2, 95% CI 1.2–4.1) and a feeling of anal blockage (OR 2.7, 95% CI 1.5–5.0) [58]. However, another study reported no difference in IBS prevalence, diagnosed on the basis of the Rome III criteria, among obese patients with and without a BE behavior pattern (*p* = 0.9) [61].

As shown, there are few data on BED and the intestine. The underlying mechanisms of lower GI symptoms in BED patients have not been clearly elucidated yet. Probably, the large amount of food, rapidly delivered in the small bowel, stimulates the intestinal secretion due to the osmotic load and promotes colonic motility, resulting in greater delivery of stool to the distal colon, reducing stool consistency and leading to diarrhea and urgency [58].

Gastrointestinal symptoms throughout the GI tract are summarized in the three main ED disorders (Figure 2).

## 6. Gut Microbiota

Deviation in the gut microbiota has been revealed in GI and extra-GI disorders, leading researchers to investigate the role of microbiota in the pathogenesis of several diseases.

The gut microbiota maintains a complex bidirectional communication system with the central nervous system (CNS) and the gut–brain Axis, involving nervous, endocrine and immune systems, contributing to the regulation of emotional behavior and cognition [112]. Interestingly, germ-free mouse models showed an exaggerated hypothalamic-pituitary-adrenal (HPA) axis response to stress [113,114], revealing that microbiota might mediate the interaction between the HPA axis, stress and immune system [115]. In addition, Gareau et al., demonstrated that chronic stress could increase intestinal permeability, allowing contact between bacteria and neuro-endocrinal cells of the ENS [116] (Figure 3).

In ED, the latest research, although often focused on AN patients [117,118,119], has demonstrated a difference in alpha-diversity and composition of microbiota compared to healthy controls [120]. For example, some authors demonstrated that gut microbiota in AN patients shows a reduction of Roseburia abundance that are involved in gut health and immune defense maintenance, primarily through the production of short-chain fatty acids (especially butyrate), possibly explaining the data about the decreased levels of short-chain fatty acids (SCFAs) in AN patients [117,118,120]. Moreover, butyrate concentrations were negatively correlated with anxiety and depression scores, suggesting in ED patients a correlation between dysbiosis and anxiety-depression disorders that has been supported by other studies [121,122]. In addition, Borgo et al. [117] showed an interesting positive correlation between decreased in *Roseburia* spp., and in particular of *Roseburia inulinivorans* and insulin levels. This hypothesis seems to propose a link between *Roseburia* spp. and insulin metabolism that helps anorectic patients preserve euglycemia [122].

Moreover, higher levels of *Enterobacteriacee* and *Methanobrevibacter smithii*, an Archaeon capable of extracting more calories from food through the transformation of hydrogen in methane [117,120,123,124], have been identified in AN compared to healthy controls. There is experimental evidence that intestinal methane production is linked to a slower intestinal transit, predisposing to constipation [125], a common AN patient’s complaint.

It has been found that the family of *Enterobacteriaceae*, in particular, *Escherichia coli* species, can produce an anorexigenic bacterial protein, the Caseinolytic protease B homologue protein (ClpB), that is an antigen mimetic of α-melanocyte-stimulating hormone (α-MSH) which triggers production of α-MSH across reactive autoantibodies. α-MSH is a hormone that plays a key role in satiety by binding to the melanocortin-4-receptor (MC4R). Recently, it has been hypothesized that the immunocomplex formed by cross-reactive autoantibodies with α-MSH chronically stimulates the melanocortine system involved in feeding behavior. It has been highlighted that ClpB may directly stimulate the production of PYY, a hormone released by enteroendocrine cells in the gut [126] involved in the regulation of appetite. It is already known that a high level of PYY is present in fasting AN patients [78].

Some authors suggested that in ED patients, a higher level of *Enterobacteriaceae* might directly activate through the release of ClpB, the satiation and satiety pathways [127,128,129]. Moreover, Breton et al. demonstrated elevated levels of ClpB in ED patients’ systemic circulation, without revealing significant differences between the ED subgroups; this evidence could suggest a pathophysiological role of this bacterial protein in ED [128].

In the last decade, several studies have analyzed the potential role of gut microbiome modulating therapy as a treatment for psychological disorders [130,131,132,133,134]. In addition to the rising knowledge about probiotic treatment for GI disorders, several studies have suggested positive effects of specific *Lactobacillus* and *Bifidobacterium* strains on mood disorders and cognitive impairment [131,135,136]. Similarly, the potential beneficial effects of prebiotics are noteworthy [137]; in fact, some evidence correlates the ingestion of certain fermentable dietary fibers with the improvement of gut microbiota composition and productive fermentation of SCFAs [131,138,139].

Recently, the prospective role of Fecal Microbiota Transplant (FMT) was evaluated in the management of ED. To date, only promising preclinical studies about the positive effects of microbiota restoration on mood disorders are available [140,141,142]; for this reason, future studies should focus on the possibility of translating these results in human studies.

Regarding the effect of weight gain on the fecal microbiota, available data are not concordant; indeed, whereas various authors agree about the increased levels of *Firmicutes*, there is no agreement regarding the amount of *Bacteroides* [118,143].

Given the suggestive hypotheses, the pathophysiological role of dysbiosis in AN should be assessed in forthcoming long-term studies.

## 7. Conclusions

ED are very complex disorders associated with high rates of morbidity and mortality.

After decades of investigation in the ED field, mechanisms underlying the abnormal behavior that characterize ED, for example eating in the absence of hunger and avoiding food in the presence of starvation, are still uncertain. GI symptoms occur very commonly in ED; however, it remains a matter of debate if GI disorders are inherent manifestations of ED or the results of malnutrition occurring in ED.

Growing interest exists in the role of intestinal microbiota in the pathogenesis of ED, in its changes through refeeding practices, and especially in the modulation of intestinal microbiota using nutritional interventions or psychobiotic application as a co-adjuvant to ED standard therapy. Although challenging for the clinicians, GI disorders in patients with ED should be investigated as well as a screening for ED encouraged in individuals seeking treatment for unexplained GI complaints. This will help to better address therapeutic issues in these difficult medical conditions. Future studies are needed to add new knowledge in this field, possibly creating new synergies among a broad array of scientists and clinicians.

## Figures and Tables

**Figure 1 nutrients-11-03038-f001:**
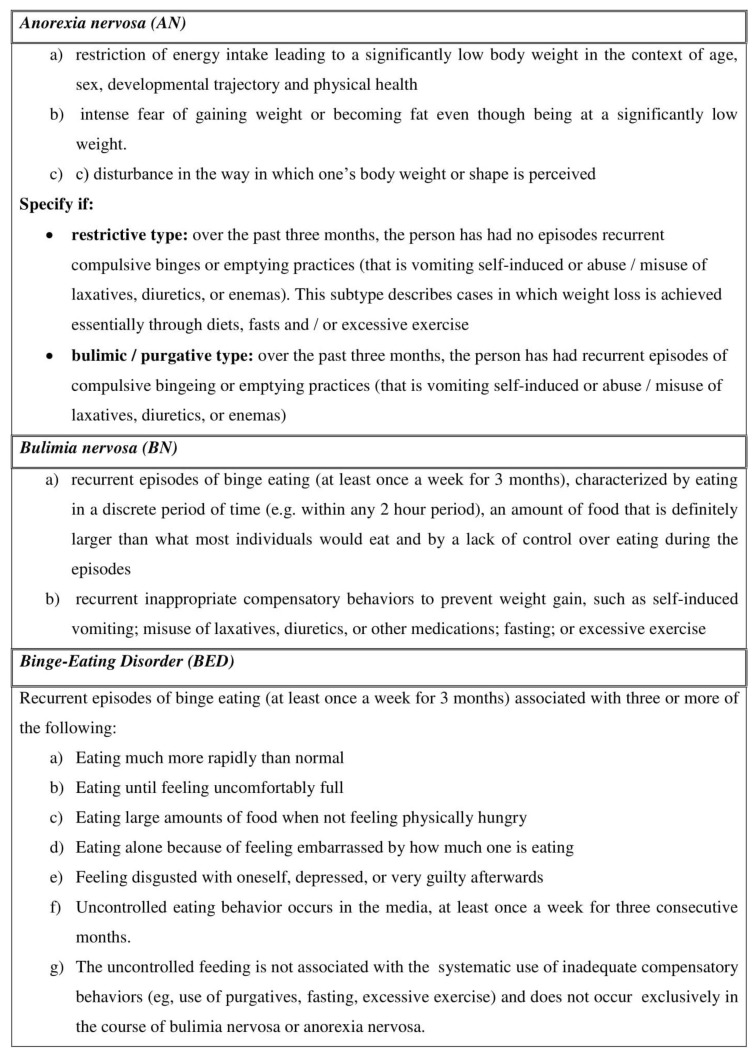
The description of the three main types of Eating Disorders (adapted by the Fifth Edition of the Diagnostic and Statistical Manual of Mental Disorders (DSM-V), released by the American Psychiatric Association [2].

**Figure 2 nutrients-11-03038-f002:**
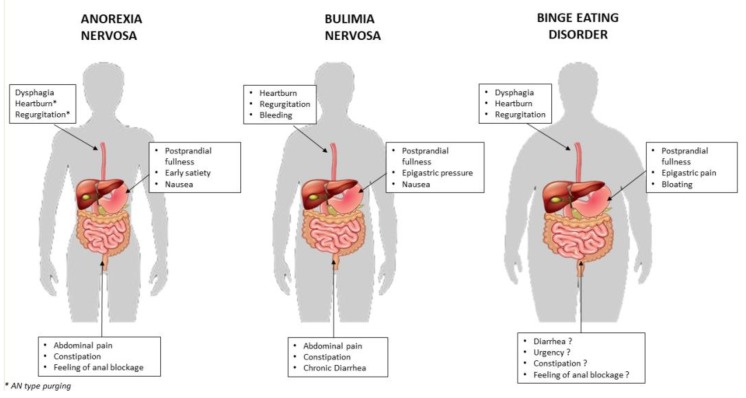
The most prevalent GI symptoms are shown taking into account different anatomical regions from esophagus to intestine in the three main ED disorders: AN, BN and BED.

**Figure 3 nutrients-11-03038-f003:**
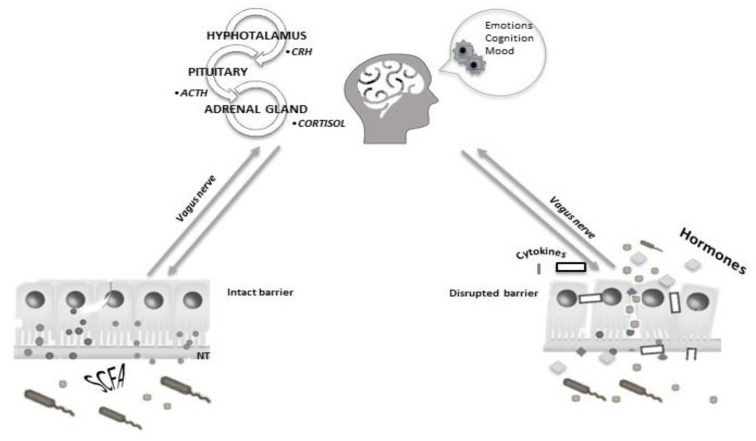
The complex bidirectional communication between the central nervous system (CNS) and microbiota involving nervous, endocrine and immune systems that might contribute to the regulation of emotional behavior and cognition in intact and disrupted intestinal barriers.

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
