# Peer review of "Eating Disorders and Gastrointestinal Diseases"

_nutrients, 2019, doi:10.3390/nu11123038_

Round 1

Reviewer 1 Report

Although of great interest to the field, this review is poorly written and organized. The different sections are unconnected and do not follow a logical order. The English language needs an important edition and, although with enormous potential, I need to recognise that this article, as it stands, does not qualify for publication in a scientific journal with an impact index. The text is not supported by figures, tables, non-significant data are shown, it is said that there are few studies, but which are not cited. In the microbiota section, although it appears in capital letters as relevant, there has been neither a recent review nor an attempt to look for the significant microbiota of all the disorders discussed (it is recommended to work on this if you want this paper to be published).

Author Response

Reviewer #1

We are grateful for the reviewer’s comments and suggestion.

Although of great interest to the field, this review is poorly written and organized. The different sections are unconnected and do not follow a logical order.

The English language needs an important edition and, although with enormous potential, I need to recognise that this article, as it stands, does not qualify for publication in a scientific journal with an impact index.

The text is not supported by figures, tables, non-significant data are shown, it is said that there are few studies, but which are not cited.

In the microbiota section, although it appears in capital letters as relevant, there has been neither a recent review nor an attempt to look for the significant microbiota of all the disorders discussed (it is recommended to work on this if you want this paper to be published).

We extensively modified the text according to reviewer’s suggestions, we added Table1 and Figure 1. The paper was completely revised by an English mother tongue. Moreover, the microbiota section was completely changed as suggested.

Reviewer 2 Report

A Santonicola et al have submitted a review entitled « Eating disorders and gastrointestinal diseases ». According to the authors, the aim of this review to « present the link between eating disorders and gastrointestinal diseases and where known the potential etiopathogenic mechanisms underlying these conditions ».

This is an interesting topic of growing interest in relation to the growing prevalence of eating disorders (ED) and the frequent comorbid condition of ED with gastrointestinal disorders (GID). Several excellent reviews covering a very similar scope have been published in the recent years and the present paper is based roughly on the same body of literature.

The paper is organized in three parts. The first is mainly dedicated to summarize the classification of ED according to DSM-5. The second and largest part of the review covers the available published data on the association of ED with disorders of esophagus, stomach and intestine. The third part briefly summarizes some of the recent studies supporting an involvement of gut microbiota in the pathophysiology of ED and may be GID. Conclusions are limited to implications for screening for ED in the presence of GID and vice-versa.

Major comment

The efforts of the authors to present an exhaustive review of the topic must be acknowledged.

Unfortunately the paper does not provide a reader-friendly synthesis of the field. The introduction should be shortened and supported by appendix data for the standard clinical description of ED. The epidemiologic data on ED and on association of ED with GID lack some updated references. The choice of the authors to organize the second part by digestive organ does not allow the reader to get a synthetic approach of digestive symptoms that can be associated to a given ED and recriprocally how digestive symptoms can reveal ED. Illustration of the data by some tables or figures would have been very helpful to the reader in the second as well as in the third part. The third part on microbiota and ED omits recent data regarding the different ways (proteins, endotoxins, metbolies…) the microbiota may use to interfere both with eating behavior regulation and gastrointestinal functions. It should also be described how well established triggering factors of ED (e.g. stress) may affect at the same time eating behaviour and digestive functions. Other comorbid conditions (anxiety, depression, addiction) are only briefly mentioned. Finally, perspectives for additional mechanistic understanding of ED and GID, for the nutritionnal modulation of the microbiota-gut-brain axis in these conditions and for future therapeutic (pharmaceutical and non pharmaceutical) approaches of associated ED-GID would have greatly enriched the interest of the paper.

Author Response

A Santonicola et al have submitted a review entitled « Eating disorders and gastrointestinal diseases ». According to the authors, the aim of this review to « present the link between eating disorders and gastrointestinal diseases and where known the potential etiopathogenic mechanisms underlying these conditions ». This is an interesting topic of growing interest in relation to the growing prevalence of eating disorders (ED) and the frequent comorbid condition of ED with gastrointestinal disorders (GID). Several excellent reviews covering a very similar scope have been published in the recent years and the present paper is based roughly on the same body of literature. The paper is organized in three parts. The first is mainly dedicated to summarize the classification of ED according to DSM-5. The second and largest part of the review covers the available published data on the association of ED with disorders of esophagus, stomach and intestine. The third part briefly summarizes some of the recent studies supporting an involvement of gut microbiota in the pathophysiology of ED and may be GID. Conclusions are limited to implications for screening for ED in the presence of GID and vice-versa. 

Major comment

The efforts of the authors to present an exhaustive review of the topic must be acknowledged.

Unfortunately the paper does not provide a reader-friendly synthesis of the field.

The introduction should be shortened and supported by appendix data for the standard clinical description of ED.

The epidemiologic data on ED and on association of ED with GID lack some updated references.

We are grateful for the reviewer’s comments and suggestions.

We modified the text adding a Table reporting the definition of the three main categories of ED: anorexia nervosa, bulimia nervosa and binge eating disorders.We also updated the references about the epidemiology of ED: the reference number 4 (Guerdjikova AI, Mori N, Casuto LS, McElroy SL. Binge Eating Disorder. The Psychiatric clinics of North America 2017; 40(2): 255-266) was changed: Udo T, Grilo CM. Prevalence and Correlates of DSM-5-Defined Eating Disorders in a Nationally Representative Sample of U.S. Adults. Biological psychiatry 2018; 84(5): 345-354

The choice of the authors to organize the second part by digestive organ does not allow the reader to get a synthetic approach of digestive symptoms that can be associated to a given ED and recriprocally how digestive symptoms can reveal ED.

We accept the reviewer’s point of view. However, we decided to illustrate the GI abnormalities associated to ED separately from the esophagus to intestine in attempt to offer a useful tool to guide physicians in their daily practice. We took into account a classification in anatomical regions because Rome Criteria adopted a similar classification for identifying clinical features (Drossman DA. Functional Gastrointestinal Disorders: History, Pathophysiology, Clinical Features and Rome IV. Gastroenterology 2016 [PMID: 27144617  DOI: 10.1053/j.gastro.2016.02.032]).

Considering that Gut Microbiota could be considered as a “new organ” inside the digestive system, we added a special section dedicated to this topic.

Illustration of the data by some tables or figures would have been very helpful to the reader in the second as well as in the third part.

The Reviewer is correct. We modified the manuscript adding a Table and a Figure

The third part on microbiota and ED omits recent data regarding the different ways (proteins, endotoxins, metbolies…) the microbiota may use to interfere both with eating behavior regulation and gastrointestinal functions.

We revised completely this section with updated references

It should also be described how well established triggering factors of ED (e.g. stress) may affect at the same time eating behaviour and digestive functions. Other comorbid conditions (anxiety, depression, addiction) are only briefly mentioned.

The Reviewer is correct. These are very interesting topics; however, considering the vastness of the issue, we do not cover triggering factors (e.g. stress, emotions) and other comorbid conditions (anxiety, depression, addiction) that may affect at the same time eating behavior and digestive function. We suggested some reviews that recently investigated about these topics (lines 94-96; page 3)

Finally, perspectives for additional mechanistic understanding of ED and GID, for the nutritionnal modulation of the microbiota-gut-brain axis in these conditions and for future therapeutic (pharmaceutical and non pharmaceutical) approaches of associated ED-GID would have greatly enriched, the interest of the paper

We agree with the reviewer and we altered the text in microbiota section adding the latest discovery in this field and speculating about future therapeutic approach.

Reviewer 3 Report

In the article „Eating disorders and gastrointestinal diseases” Santonicola et al. review the present knowledge about the impact of eating disorders on gastrointestinal functioning. The following points need to be addressed:

In the abstract the section describing the aim and conclusion of the paper has a different style than the rest of the abstract. In the abstract it says the GI tract is the dominant organ involved in food intake and its regulation; however, this is too simple, since brain areas are strongly associated with these processes, too. Please revise. Please specify the statement “highly relevant” in ll. 30. Please improve the formatting in the introduction to be more consistent. The sentence “In fact, BED men more frequently complain of neck‐shoulder pain, low back pain, and chronic muscular pain, and report impairment due to physical health problems, after adjustment for body mass index“ is not relevant and should be removed. The paragraph “Eating disorders and GI symptoms” is a simple listing of findings. They are not put into context, analyzed, evaluated; no gain of knowledge is obtained. This should be completely revised. The beginning of the section „ED and esophagus” is very similar to parts of the systematic review by Schalla et al. 2019 „Gastrointestinal alterations in Anorexia Nervosa – A systematic review”. This could be omitted by focusing on BN or BED. To fulfill the characteristics of a valuable review the authors should address gaps in knowledge and introduce explanations for contradicting results like it was done in ll.151-153 and ll. 226-232 and ll.315-316 throughout the whole manuscript. The section from ll.187 to ll.203 should be shortened or in parts relocated as background information to the findings presented below. The same applies to the ll.328-348. Since CCK and PYY are not secreted from the stomach, the findings about those peptides should be removed from the section “ED and stomach”. Information given in ll.264 to ll.268 are repetitions and do not fit this section. The whole section “microbiota and eating disorders” is very similar to parts of the above-mentioned systematic review, no new literature was included. Please extend or remove. The review would benefit from any kind of illustration.

Author Response

We appreciated the reviewer’s comments and suggestions

In the article „Eating disorders and gastrointestinal diseases” Santonicola et al. review the present knowledge about the impact of eating disorders on gastrointestinal functioning. The following points need to be addressed:

In the abstract the section describing the aim and conclusion of the paper has a different style than the rest of the abstract. In the abstract it says the GI tract is the dominant organ involved in food intake and its regulation; however, this is too simple, since brain areas are strongly associated with these processes, too. Please revise.

We agree with Review suggestions and modify the abstract accordingly

Please specify the statement “highly relevant” in ll. 30.

In this version text was amended Line  29-32 page 1

Please improve the formatting in the introduction to be more consistent. The sentence “In fact, BED men more frequently complain of neck‐shoulder pain, low back pain, and chronic muscular pain, and report impairment due to physical health problems, after adjustment for body mass index“ is not relevant and should be removed.

In this version we deleted the sentence

The paragraph “Eating disorders and GI symptoms” is a simple listing of findings. They are not put into context, analyzed, evaluated; no gain of knowledge is obtained. This should be completely revised.

We completely deleted the paragraph and reported the symptoms in the respective anatomical regions

The beginning of the section „ED and esophagus” is very similar to parts of the systematic review by Schalla et al. 2019 „Gastrointestinal alterations in Anorexia Nervosa – A systematic review”. This could be omitted by focusing on BN or BED.

As described in our aim (Pg3, Line 102-104):

Our aim was to systematically describe GI involvement from esophagus to intestine in the three main ED disorders AN, BN and BED in an attempt to offer a useful tool to practicing physicians in the recognition and, possibly, in planning the management of GI alterations in these difficult patients.

However, we amended the text of ED and esophagus in AN paragraph.

To fulfill the characteristics of a valuable review the authors should address gaps in knowledge and introduce explanations for contradicting results like it was done in ll.151-153 and ll. 226-232 and ll.315-316 throughout the whole manuscript.

We agree with the reviewer and we modify the text accordingly.

The section from ll.187 to ll.203 should be shortened or in parts relocated as background information to the findings presented below. The same applies to the ll.328-348. Information given in ll.264 to ll.268 are repetitions and do not fit this section.

We accept the suggestion of the Reviewer and modify the text accordingly.

Since CCK and PYY are not secreted from the stomach, the findings about those peptides should be removed from the section “ED and stomach”.

The Review is correct. However, we believe that the role of CCK and PYY on satiety is better explained in the stomach section together with other hormone such as ghrelin (Text not amended).

The whole section “microbiota and eating disorders” is very similar to parts of the above-mentioned systematic review, no new literature was included. Please extend or remove.

In this version the microbiota section was completely changed adding the latest discovery in this field and speculating about future therapeutic approach.

The review would benefit from any kind of illustration. 

We accept the suggestion of the Reviewer and added Table 1 and Figure 1.

Round 2

Reviewer 1 Report

I have previously rejected this article because of the big deficiencies detected during my review. I cannot find the rebuttal letter to my comments so I am rejecting again. Now I found the same article with small modifications but with huge mistakes in scientific terms. There is an absolute lack of consensus in the use of microorganism taxonomy (I cannot believe that a paper can be published with things as E. Coli (this is not scientific nomenclature). The first time you include the terms you must use cursive and the complete Latin name: Escherichia coli (E.coli) and then later E. coli. The same happens with species, genus, Family that are used as it were a 1st-year-grade student. In addition to that, this is a scientific review without methodology, without tables, images, ... 

There are many mistakes when listing the bibliography:

1. 1 Lewinsohn PM, Hops H, Roberts RE, Seeley JR, Andrews JA. Adolescent psychopathology: I. 389 Prevalence and incidence of depression and other DSM-III-R disorders in high school students. Journal of 390 abnormal psychology 1993; 102(1): 133-144 [PMID: 8436689] 391
2. 2 Association

Author Response

Reviewer 1

"I have previously rejected this article because of the big deficiencies detected during my review. I cannot find the rebuttal letter to my comments so I am rejecting again."

We appreciated the reviewer’s comments and suggestions.

Unfortunately there has been a misunderstanding because we uploaded a letter to answer the points that you previously raised and it is reported here:

“We are grateful for the reviewer’s comments and suggestion.

We extensively modified the text according to reviewer’s suggestions; we added Table1 and Figure 1. The paper was completely revised by an English mother tongue. Moreover, the microbiota section was completely changed as suggested

"Now I found the same article with small modifications"

in the last revision the microbiota section was completely changed as suggested.

In this version, in the microbiota section text was amended again.

"but with huge mistakes in scientific terms. There is an absolute lack of consensus in the use of microorganism taxonomy (I cannot believe that a paper can be published with things as E. Coli (this is not scientific nomenclature). The first time you include the terms you must use cursive and the complete Latin name: Escherichia coli (E.coli) and then later E. coli. The same happens with species, genus, Family that are used as it were a 1st-year-grade student."

The Reviewer is correct, we changed the microorganism taxonomy following  his/her suggestions.

"In addition to that, this is a scientific review without methodology, without tables, images, ..."

In the last revision we already added a Table 1 and Figure 1, and in this version we added also the Figure 2.

"There are many mistakes when listing the bibliography:

1 Lewinsohn PM, Hops H, Roberts RE, Seeley JR, Andrews JA. Adolescent psychopathology: I. 389 Prevalence and incidence of depression and other DSM-III-R disorders in high school students. Journal of 390 abnormal psychology 1993; 102(1): 133-144 [PMID: 8436689] 391"

We completely revised also the bibliography.

Reviewer 2 Report

The authors responded adequately to most of the reviews comments.

Author Response

"The authors responded adequately to most of the reviews comments."

We have a deep appreciation for reviewer’s suggestions that have made it possible to achieve this result.

Reviewer 3 Report

I recommended to reject the paper already.

Author Response

"I recommended to reject the paper already."

 We have previously performed a deep revision following his/her suggestions and, we strongly believed that our manuscript is improved with your help.
